# Resident macrophages acquire innate immune memory in staphylococcal skin infection

Reinhild Feuerstein[1†], Aaron James Forde[1,2†], Florens Lohrmann[1,3,4], Julia Kolter[1], Neftali Jose Ramirez[1], Jakob Zimmermann[5], Mercedes Gomez de Agüero[5], Philipp Henneke[1,3]*

[1]Institute for Immunodeficiency, Center for Chronic Immunodeficiency, Medical Center, Faculty of Medicine, University of Freiburg, Freiburg, Germany; [2]Faculty of Biology, University of Freiburg, Freiburg, Germany; [3]Center for Pediatrics and Adolescent Medicine, Medical Center, Faculty of Medicine, University of Freiburg, Freiburg, Germany; [4]Spemann Graduate School of Biology and Medicine (SGBM), Faculty of Biology, University of Freiburg and IMM-PACT Clinician Scientist Program, Faculty of Medicine, University of Freiburg, Freiburg, Germany; [5]Maurice Müller Laboratories (Department for Biomedical Research), Universitätsklinik für Viszerale Chirurgie und Medizin Inselspital, University of Bern, Bern, Switzerland

*For correspondence:
philipp.henneke@uniklinik-freiburg.de

[†]These authors contributed equally to this work

Competing interests: The authors declare that no competing interests exist.

**Abstract** *Staphylococcus aureus (S. aureus)* is a common colonizer of healthy skin and mucous membranes. At the same time, *S. aureus* is the most frequent cause of skin and soft tissue infections. Dermal macrophages (Mφ) are critical for the coordinated defense against invading *S. aureus,* yet they have a limited life span with replacement by bone marrow derived monocytes. It is currently poorly understood whether localized *S. aureus* skin infections persistently alter the resident Mφ subset composition and resistance to a subsequent infection. In a strictly dermal infection model we found that mice, which were previously infected with *S. aureus*, showed faster monocyte recruitment, increased bacterial killing and improved healing upon a secondary infection. However, skin infection decreased Mφ half-life, thereby limiting the duration of memory. In summary, resident dermal Mφ are programmed locally, independently of bone marrow-derived monocytes during staphylococcal skin infection leading to transiently increased resistance against a second infection.

## Introduction

S*taphylococcus aureus* (*S. aureus*) is an important human pathogen, which causes severe invasive infections including osteomyelitis, endocarditis, pneumonia, and septicemia (*Noskin et al., 2007*). On the other hand, *S. aureus* stably colonizes the anterior nares and moist areas of the skin in up to 30% of healthy people with a high fidelity of clones (*Verhoeven et al., 2014*). Accordingly, *S. aureus* and its individual human host usually maintain a stable partnership with high adaptation at the muco-cutaneous membranes. Specific and lasting immunity to virulent microorganisms has traditionally been assigned to the adaptive immune system, more specifically to long-lived memory T cells and B cells (*Ahmed et al., 2009*). However, in the case of *S. aureus*, high frequencies of *S. aureus*-specific antibodies and circulating T cells do not safely protect from invasive infections (*Stentzel et al., 2015*). Moreover, humans and mice with global defects in lymphocyte development do not exhibit an exquisite susceptibility to *S. aureus* infection, despite the frequent encounters outlined above (*Schmaler et al., 2011*). In contrast, infections with *S. aureus* are a primary indicator of a deficiency

in cellular innate immunity, that is MyD88/IRAK4 deficiency and chronic granulomatous disease (*Feuerstein et al., 2017*).

Accordingly, several studies have identified Mφ as being essential for the rapid and coordinated defense against *S. aureus* invading the skin (*Feuerstein et al., 2015*; *Abtin et al., 2014*). The concept of innate immune memory, where microbial effectors induce mechanisms in mononuclear phagocytes which increase their ability to protect against infections, offers an attractive solution to this conundrum (*Quintin et al., 2012*; *Chan et al., 2017*; *Schrum et al., 2018*; *Yoshida et al., 2015*; *Kleinnijenhuis et al., 2012*; *Cheng et al., 2014*; *Netea et al., 2011*). As an example, priming with LPS boosts resistance to *S. aureus* 3 weeks after injection (*Yoshida et al., 2015*). More specifically, it was very recently reported that staphylococcal soft tissue infections induce a more robust immune response to a second challenge (*Chan et al., 2018*). Mechanisms for the establishment of innate memory involve phosphorylation of the stress-response transcription factor ATF7, epigenetic programming through histone modifications leading to stronger gene transcription upon re-stimulation as well as metabolic changes of Mφ via mTOR- and HIF-1α–mediated aerobic glycolysis (*Yoshida et al., 2015*; *Cheng et al., 2014*). These observations have indeed largely reshaped our understanding of immunological memory, which can currently best be conceptualized as multidimensional and gradual (*Pradeu and Du Pasquier, 2018*). Despite the increasing consensus that immunological memory in myeloid cells impacts on host defense, its contribution to infections in specific compartments remains unclear. As an example, certain organisms, for example mycobacteria, appear to alter transcriptional programs on the stem cell level (*Kaufmann et al., 2018*). This seems particularly important for the memory of tissue Mφ resident in submucous tissues like the dermis, which are seeded prenatally and are then rapidly replaced by monocyte-derived Mφ. In staphylococcal skin infection, the dermal Mφ subset composition changes due to infiltrating Ly6C$^{hi}$ monocytes (*Feuerstein et al., 2015*). Scrutinizing whether, and if so how, these dynamic alterations impact on the memory response of the individual Mφ and of the skin as an organ seems essential to unravel target cells and genes and to ultimately modify local host resistance. Thus, we have analyzed in an intradermal *S. aureus* infection model, whether, in which cells and how an '*S. aureus* memory status' develops and how this impacts on Mφ subset composition in the dermis, as well as on the response to a second infection with the same pathogen in vivo.

We found that staphylococcal skin infection directly induced a memory signature in individual dermal Mφ, signified by the increased expression of STAT1 and CXCL9, but at the same time increased Mφ turnover by naive bone-marrow derived cells, which eventually extinguished the memory. Since colonization only partially induced memory, it is tempting to speculate on minor penetrating infections to be a prerequisite for an efficient and lasting local secondary response.

## Results

### Intradermal *S. aureus* infection increases resistance to secondary infection

In order to analyze the impact of an *S. aureus* infection on immunity against a second infection, mice were intradermally (i.d.) infected with *S. aureus* (primed), were left untreated or received a vehicle injection. After 3 weeks, the mice were (re-)infected with the same inoculum (*Figure 1A* and *Figure 1—figure supplement 1C*). Bacterial burden and myeloid cell composition of the skin were analyzed after 5 days of (re-)infection, when initial neutrophil (PML) recruitment ceased and the bacterial load was declining (*Feuerstein et al., 2015*). 5 days after infection incoming Ly6C$^{hi}$ dermal Mφ can be distinguished from resident Ly6C$^{lo}$ cells (within the CD64$^{hi}$ population) based on a 'waterfall' of Ly6C and MHC2 expression (*Figure 1—figure supplement 1B*). 3 weeks after primary *S. aureus* infection, that is immediately before the second infection, CD64$^{hi}$ dermal Mφ (gating strategy depicted in *Figure 1—figure supplement 1A*) were increased (*Figure 1C*), whereas *S. aureus* could not be cultured and PML were virtually absent (*Figure 1H*). Previous infection with *S. aureus* (primed) significantly increased bacterial killing and was associated with fewer neutrophils in the tissue 5 days after the 2$^{nd}$ infection (*Figure 1B*, *Figure 1D*). Next, we determined the kinetics of dermal myeloid cell recruitment during secondary infection with *S. aureus* in naïve and primed mice at five dpi. In addition to the reduced number of neutrophils, primed mice had less Ly6C$^{hi}$ skin macrophages compared to naïve mice (*Figure 1E*). As shown in our previous work, Ly6C$^{hi}$ CD64$^{hi}$ CX$_3$CR1$^{int}$ skin

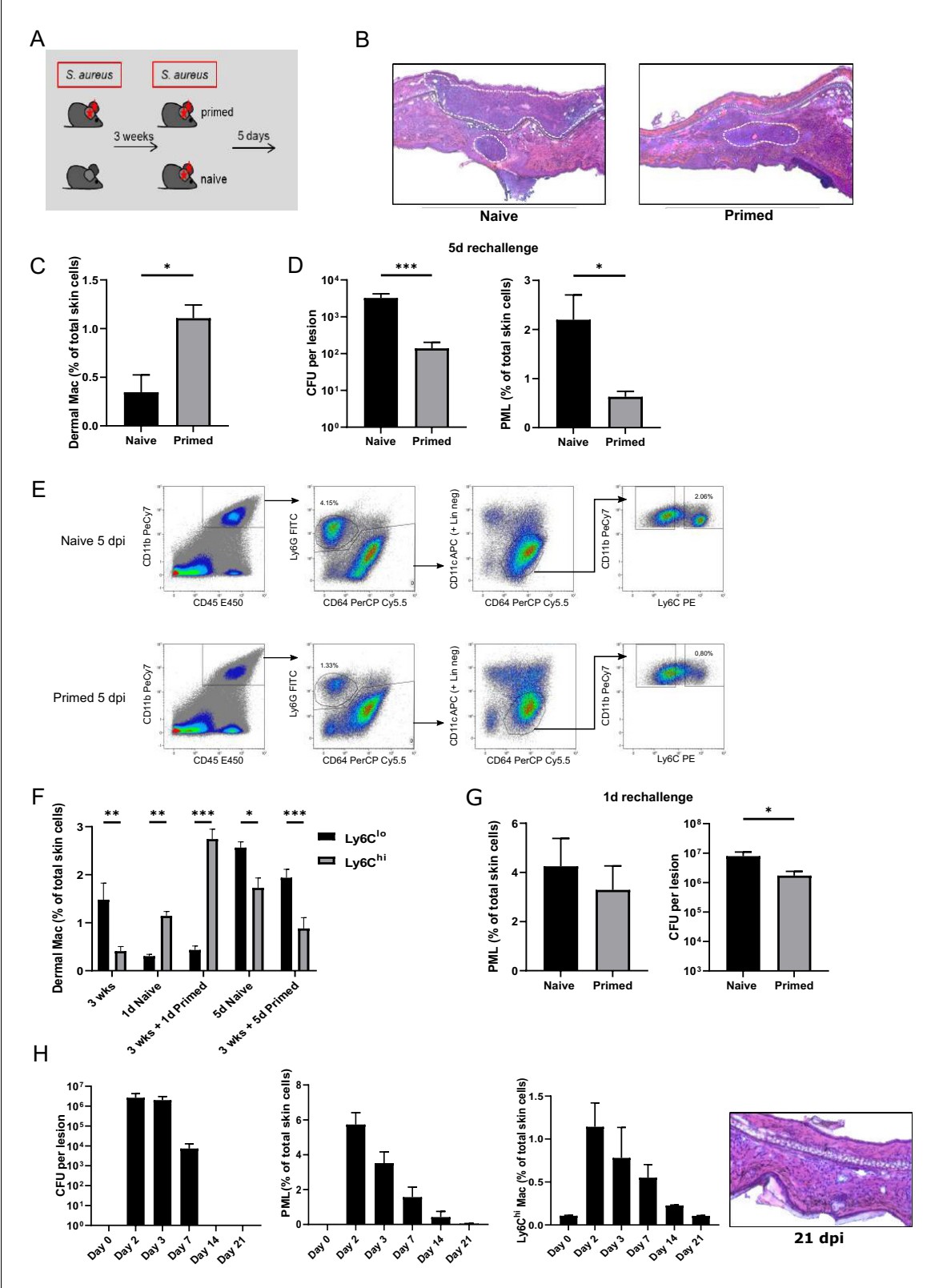

**Figure 1.** Intradermal *S. aureus* infection increases resistance to secondary infection. (A, D) Wt mice were i.d. infected with *S. aureus* (10$^7$ CFU in 10 µl PBS (primed) or not (naive)), (re-)infected 3 weeks later with the same bacterial dosage and analyzed for bacterial load and skin infiltrating PML after an additional 5 days (n: eight biological replicates representing two independent experiments). (B) HE staining of cryosections from naïve and previously infected mice 5 days after (re-)infection. Dashed lines mark immune cell infiltrations. (C) Quantification of dermal Mφ in wt mice 3 weeks after primary

*Figure 1 continued on next page*

*Figure 1 continued*

infection and without secondary infection, data is gated on all CD64$^{hi}$ macrophages. (n: three biological replicates representing three independent experiments). (E) Representative FACS plots of the dermal myeloid cell composition in naïve and primed mice treated as in (A) 5 days after secondary infection (numbers represent % of total skin cells). (F) Frequency of Ly6C$^{lo}$ and Ly6C$^{hi}$ dermal Mφ 3 weeks after primary infection as well as after (re-)infection periods of 1 and 5 days in primed mice and mice infected for the first time (naïve). (G) Wt mice infected with *S. aureus* (10$^7$ CFU in 10 µl PBS) and (re-)infected 3 weeks later (primed) or for the first time (naïve) with the same amount of bacteria and analyzed for bacterial load and skin infiltrating PML after (re-)infection for 1 day (n: 7–8 biological replicates representing two independent experiments). (H) CFU counts, skin neutrophil and Ly6C$^{hi}$ dermal Mφ numbers of WT mice up to 21 days after i.d. *S. aureus* infection and without secondary infection. HE staining of cryosections from wt mice 3 weeks after i.d. *S. aureus* infection. Data were analyzed using either two tailed Mann-Whitney test or two-way ANOVA with Bonferroni's multiple comparison test. Error bars are mean ± SEM. *p<0.05, **p<0.01, ***p<0.001.

The online version of this article includes the following figure supplement(s) for figure 1:

**Figure supplement 1.** Gating Strategy used to distinguish dermal macrophages.

Mφ are progeny of recruited Ly6C$^{hi}$ monocytes and will eventually differentiate into resident Ly6C$^{lo}$ Mφ (*Feuerstein et al., 2015*; *Kolter et al., 2019*). In previously infected mice, we observed a slight increase of Ly6C$^{lo}$ dermal Mφ and an increase of Ly6C$^{hi}$ dermal Mφ already 24 hr after (re-)infection while by day 5 Ly6C$^{hi}$ dermal Mφ were lower in primed mice (*Figure 1E*, *Figure 1F*). In contrast to Ly6C$^{hi}$ dermal Mφ, PML counts were largely unaffected one day after secondary infection. However, increased bacterial killing was already apparent one day after reinfection (*Figure 1G*). To further explore, whether the enhanced antimicrobial properties after (re-)infection were due to ongoing inflammation after the primary infection or rather recall phenomena, we analyzed bacterial load, neutrophil counts and histological alterations for three weeks after primary infection. We found that bacteria were no longer detectable and that skin infiltrating PML, Ly6C expression and tissue edema were back to base line 2 weeks after infection (*Figure 1H*).

## Infection-induced resistance against *S. aureus* does not depend on bone marrow-derived monocytes and on mature T, B, and NK cells

To investigate the role of bone marrow-derived cells in the establishment of induced resistance against *S. aureus,* we analyzed the proportion of Ly6C$^{hi}$ Mφ, which are putative monocyte progeny (*Kolter et al., 2019*), and of Ly6C$^{lo}$ dermal Mφ during (re-)infection. During steady state, Ly6C$^{hi}$ dermal Mφ only accounted for less than 1% of skin immune cells. After secondary infection periods of 1 or 5 days, the number of Ly6C$^{hi}$ dermal Mφ increased to 2–3% of total skin cells in wt mice (*Figure 1F*). While this increase was absent in *Ccr2$^{-/-}$* mice, Ly6C$^{lo}$ Mφ numbers were similar to wt mice (*Figure 2A* and *Figure 2—figure supplement 1A*). This indicates a dynamic turnover process which is highly dependent on the respective timeline of infection. We next tested whether, despite significantly reduced numbers of inflammatory monocytes, enhanced resistance against *S. aureus* could be induced in *Ccr2$^{-/-}$* mice in vivo. First, we found naïve *Ccr2$^{-/-}$* mice to exhibit an overall higher bacterial burden after primary *S. aureus* infection. However, a rechallenge of previously infected *Ccr2$^{-/-}$* mice revealed increased bacterial killing (as indicated by a reduction in *S. aureus* CFU), decreased infiltrating PML, and less skin IL-1β concentrations as compared to naïve counterparts 5 days after (re-)infection (*Figure 2B*). To further investigate whether this memory phenotype could be transferred with bone marrow from primed to naïve mice, we injected CD45.1 mice i.d. with *S. aureus* or PBS. After 2 weeks, bone marrow from these mice was transferred into lethally irradiated naïve wt recipient mice. After an additional 2 weeks, recipient mice were infected with *S. aureus* for 5 days. In concordance with our previous finding that monocytes were not essential for innate memory after dermal *S. aureus* infection, we did not observe a significant difference in bacterial killing and neutrophil recruitment between recipient mice, which were reconstituted with bone marrow from infected mice, and those which were reconstituted with bone marrow from PBS treated mice (*Figure 2—figure supplement 1B*).

Next we determined, whether *S. aureus*-induced immune memory was dependent on the adaptive immune system. Therefore, we analyzed *Rag2$^{-/-}$γc$^{-/-}$* mice, which lack mature B, T and NK cells, but have similar skin Mφ numbers compared to wt mice (*Figure 2D* and *Figure 1F*), for enhanced protection against bacteria by re-exposure. Primed *Rag2$^{-/-}$γc$^{-/-}$* mice showed a clear memory phenotype with increased bacterial killing, reduced PML and IL-1β tissue concentration at 5 days p.i. (*Figure 2C*). Therefore, the lasting protective effects exerted by *S. aureus* did not depend on bone-

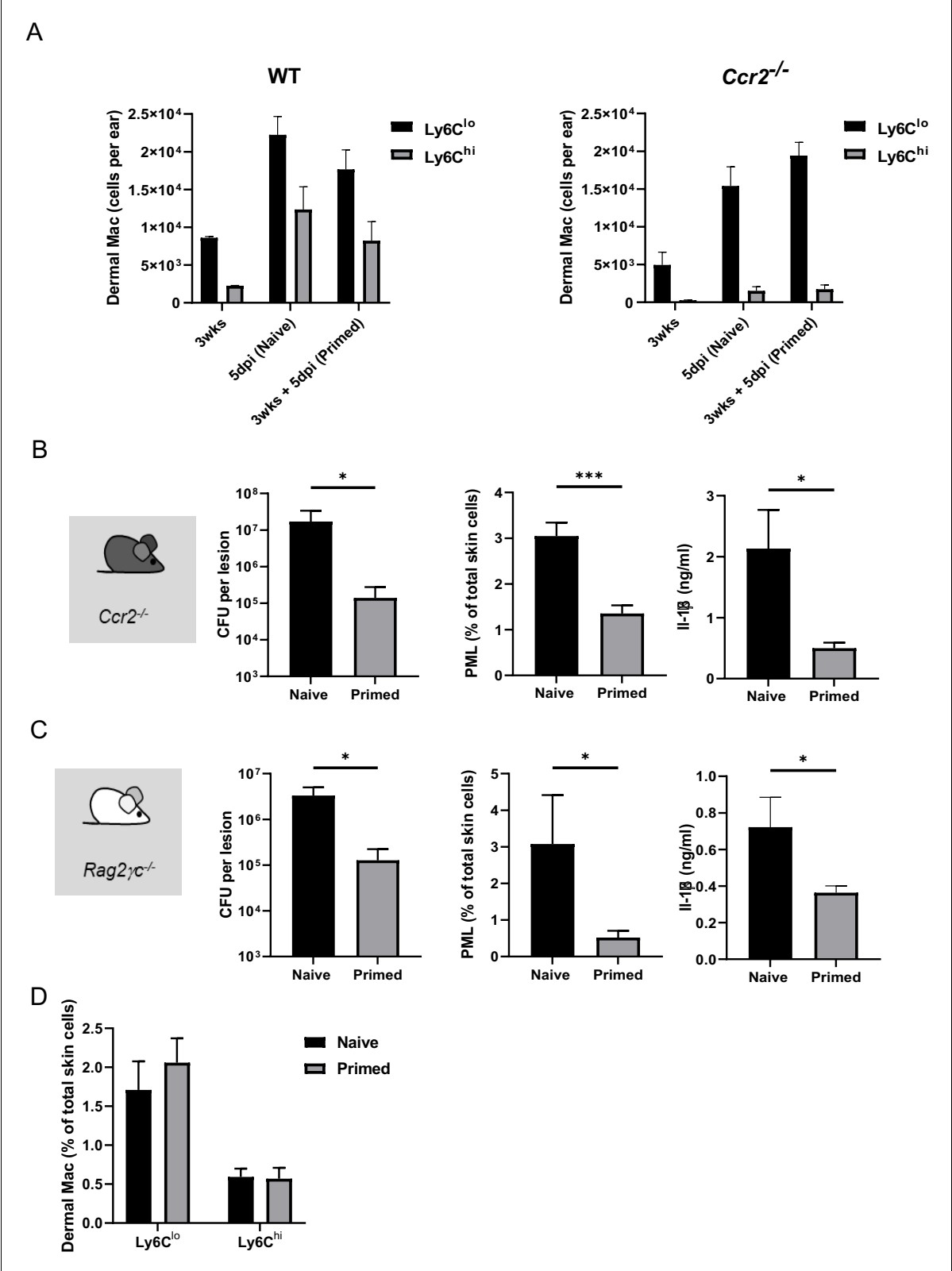

**Figure 2.** Innate memory does not depend on bone marrow-derived monocytes and on mature T, B, and NK cells. (**A**) Quantification of Ly6C[hi] and Ly6C[lo] dermal Mφ in mice infected 3 weeks previously, naïve mice infected for the first time and primed mice, 5 days after i.d. (re-)infection with *S. aureus* in wt (left panel) and *Ccr2*[-/-] mice (right panel). (**B**) *Ccr2*[-/-] mice were i.d. infected with *S. aureus* (10[7] CFU in 10 μl PBS) and i.d. (re-)infected 3 weeks later (primed) or for the first time (naïve) with the same amount of bacteria and analyzed for bacterial load, skin infiltrating PMLs and skin IL-1β

*Figure 2 continued on next page*

*Figure 2 continued*

levels after additional 5 days (n: 11 biological replicates representing three independent experiments). (C) *Rag2⁻/⁻γc⁻/⁻* mice were i.d. infected with *S. aureus* ($10^7$ CFU in 10 µl PBS) and i.d. (re-)infected 3 weeks later (primed) or for the first time (naïve) with the same amount of bacteria and analyzed for bacterial load, skin infiltrating PML and skin IL-1β levels after additional 5 days (n: 5–7 biological replicates from two independent experiments). (D) Quantification of Ly6C^hi and Ly6C^lo dermal Mφ in naïve and primed *Rag2⁻/⁻γc⁻/⁻* mice 5 days after (re-)infection. Data were analyzed using two tailed Mann-Whitney test. Error bars are mean ± SEM. *p<0.05, ***p<0.001.

The online version of this article includes the following figure supplement(s) for figure 2:

**Figure supplement 1.** Bone marrow cells do not contribute to innate memory.

marrow derived monocytes and the adaptive immune system and qualified as 'innate immune memory'.

## Resident dermal M$\varphi$ mediate innate memory

To further demonstrate that dermal Mφ were responsible for the enhanced resistance to secondary *S. aureus* infection, we treated previously infected mice with either liposomal clodronate or control liposomes (*Figure 3A*). This method depletes about 70–80% of all CD64^hi dermal Mφ within 2 days by induction of apoptosis (*Feuerstein et al., 2015*). Repopulation of dermal Mφ occurred one week after depletion (*Figure 3B*). We injected clodronate liposomes into mice 2.5 weeks after *S. aureus* infection to deplete resident dermal Mφ. After an additional 1.5 weeks mice were (re-)infected. We found reduced bacterial killing, increased PML numbers and higher IL-1β concentrations in the skin

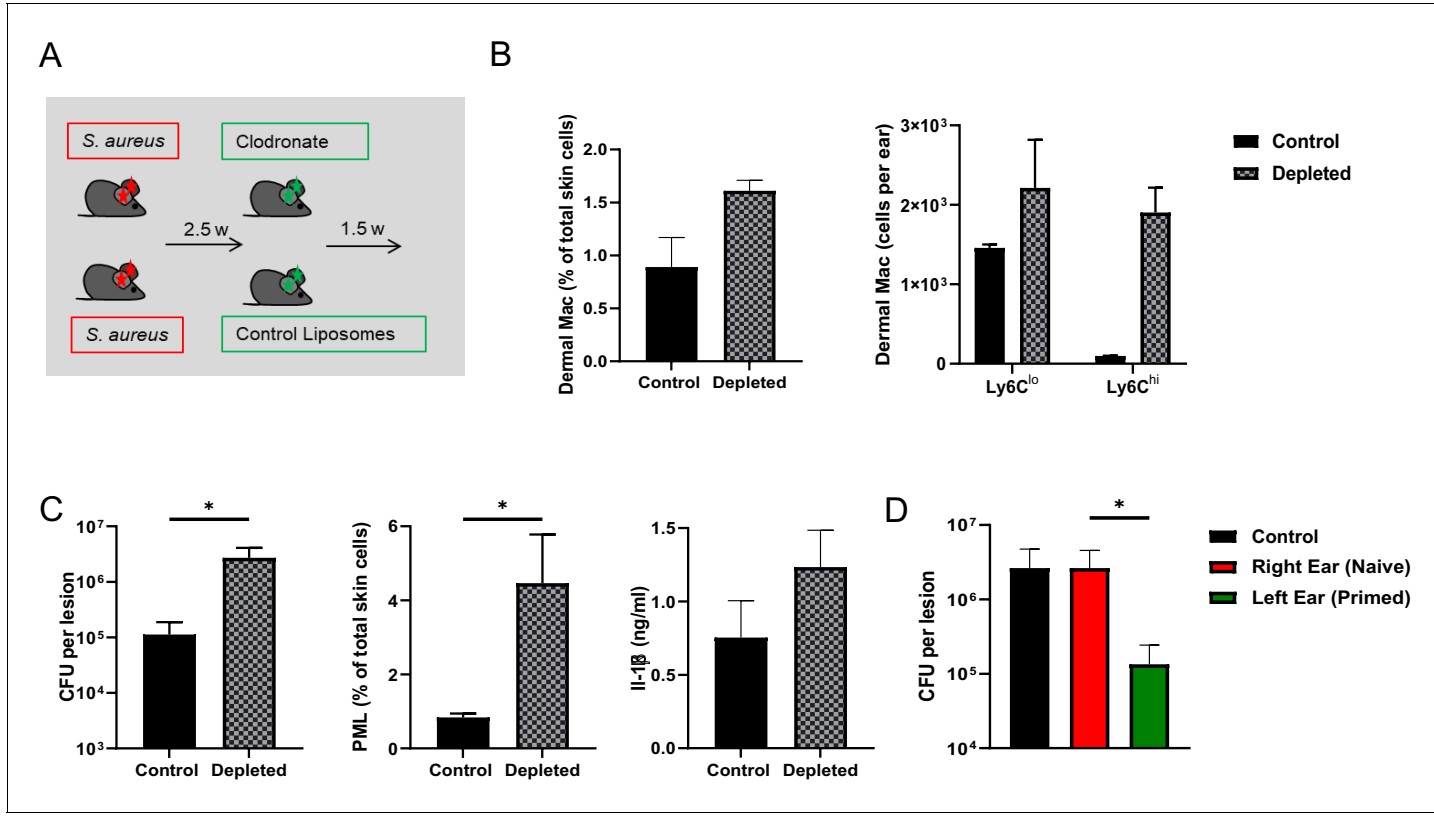

**Figure 3.** Dermal Mφ mediate innate memory. (A) Wt mice were i.d.infected with *S. aureus* ($10^7$ CFU in 10 µl PBS) and 2.5 weeks later i.d. injected with either clodronate liposomes or control liposomes. (B) Quantification of CD64^hi, Ly6C^hi and Ly6C^lo dermal Mφ from control and depleted mice treated as in (A). (C) Wt mice were treated as in (A) and re-challenged with *S. aureus* ($10^7$ CFU in 10 µl PBS) 1.5 weeks after clodronate treatment and analyzed for bacterial load, skin infiltrating PML and skin IL-1β levels after 5 days (n: eight biological replicates from two independent experiments). (D) Only the left ears of wt mice were i.d. infected with *S. aureus* for 3 weeks and the memory status of the corresponding right ear analyzed after additional infection of both ears for 5 days (n: 10–11 biological replicates from three independent experiments). Data were analyzed using two tailed Mann-Whitney test. Error bars are mean ± SEM. *p<0.05. (in D the groups right ear and left ear were compared).

of clodronate treated mice (*Figure 3C*). This indicated that depletion of the resident dermal Mφ pool abolished innate memory, although the Mφ density was maintained due to invasion of monocytes. We next tested, whether infection of contralateral tissue (ear) could increase the protection as well. Interestingly, innate memory could only be observed after a re-challenge of the same ear (*Figure 3D*), indicating that it was primarily mediated by local resident Mφ without contribution of blood- or bone marrow-derived cells.

## Innate memory alters the transcriptional signature in dermal M$\varphi$

We next investigated the transcriptional changes associated with innate memory in dermal Mφ. Resident dermal Mφ were sorted from *S. aureus* infected and control $Ccr2^{-/-}$ mice, which were used in order to specifically analyze the memory program in resident dermal Mφ, which – as shown above - essentially mediate increased resistance to a secondary infection. Three weeks after *S. aureus* infection, 65 genes were differentially expressed as compared to Mφ from vehicle control mice including the transcription factor *Stat1* (*Figure 4A*). Gene ontology analysis using the Panther bioinformatics platform revealed enrichment of terms associated with antigen presentation, response to interferons and positive regulation of cytokine production (*Figure 4B*). Accordingly, intradermal *S. aureus* infection induced lasting changes in a distinct subset of genes which was associated with increased resistance to secondary infection.

## Dermal M$\varphi$ show a specific memory signature and increased responsiveness after re-infection ex vivo

Upregulation of the transcription factor *Stat1*, as observed in the gene expression array, and its target *Cxcl9*, was confirmed by quantitative PCR to be increased in sorted CD64$^{hi}$ Mφ from wt mice previously infected with *S. aureus* as compared to naïve mice (*Figure 5A and B*). These genes have been shown to be involved in innate memory in LPS-primed Mφ, which also show an enhanced resistance to *S. aureus* after 3 weeks (*Yoshida et al., 2015*). Accordingly, phosphorylated STAT1 was increased in dermal Mφ from whole tissue lysates of *S. aureus* infected skin (*Figure 5C*). Furthermore, Mφ of previously infected mice had increased expression of *Stat1* and *Cxcl9* 1 and 5 days after (re-)infection (*Figure 5D*). In addition, increased concentrations of CXCL9 were detectable 5 days after (re-)infection in skin lysates of primed mice compared to naïve controls (*Figure 5E*). Notably, CXCL9 has been reported to exhibit direct antimicrobial activity independent of its chemotactic activity (*Reid-Yu et al., 2015*). We confirmed this observation by an in vitro *S. aureus* killing assay showing the direct killing capacity of CXCL9 on the *S. aureus* strain Newman (*Figure 5—figure supplement 1*). Notably, in dermal Mφ sorted from primed mice that had received clodronate (as in *Figure 3A*), the enhanced expression of *Stat1* and *Cxcl9* was lost (*Figure 5—figure supplement 2A*).

We then analyzed the response of previously infected dermal Mφ to a second challenge with *S. aureus* ex vivo. Accordingly, CD64$^{hi}$ dermal Mφ were sorted from mice 21 days after *S. aureus* infection or from naive mice and then stimulated ex vivo with fixed *S. aureus*. We found that dermal Mφ from primed mice showed higher pro *IL1B* expression and more TNF-α production than Mφ from naïve mice (*Figure 5F*). This finding suggested that in vivo priming leads to increased responsiveness to the same pathogen at the single Mφ level.

We have previously shown that rather subtle differences in eliciting inflammation in the earliest stages of staphylococcal infection translate into altered skin immunity for days. In these early stages, the Mφ response in vivo is similar regardless of whether live or fixed *S. aureus* is used (*Feuerstein et al., 2015*). Thus, the early Mφ response relies on recognition of the bacterial particle, rather than on secreted factors or active proliferation. The usage of fixed bacteria allows a focus on the Mφ activation status irrespective of the bacterial killing capacity. Upon challenge of primed mice for 4 hr with fixed *S. aureus* (*Figure 5G*), we observed significantly more skin-infiltrating PML and Ly6C$^{hi}$ Mφ (*Figure 5H*) and visible skin erythema in previously infected mice (*Figure 5—figure supplement 2B*). In full agreement with the above results, primed mice had increased concentrations of CXCL1 in whole tissue lysates compared to naïve counterparts (*Figure 5I*). This strongly suggested that the initial response to *S. aureus* primed Mφ for an increased responsiveness to the second infection resulting in the more rapid recruitment of peripheral blood cells. While enhanced bacterial killing occurs in the absence of inflammatory monocytes (*Figure 2B*) we aimed to investigate whether

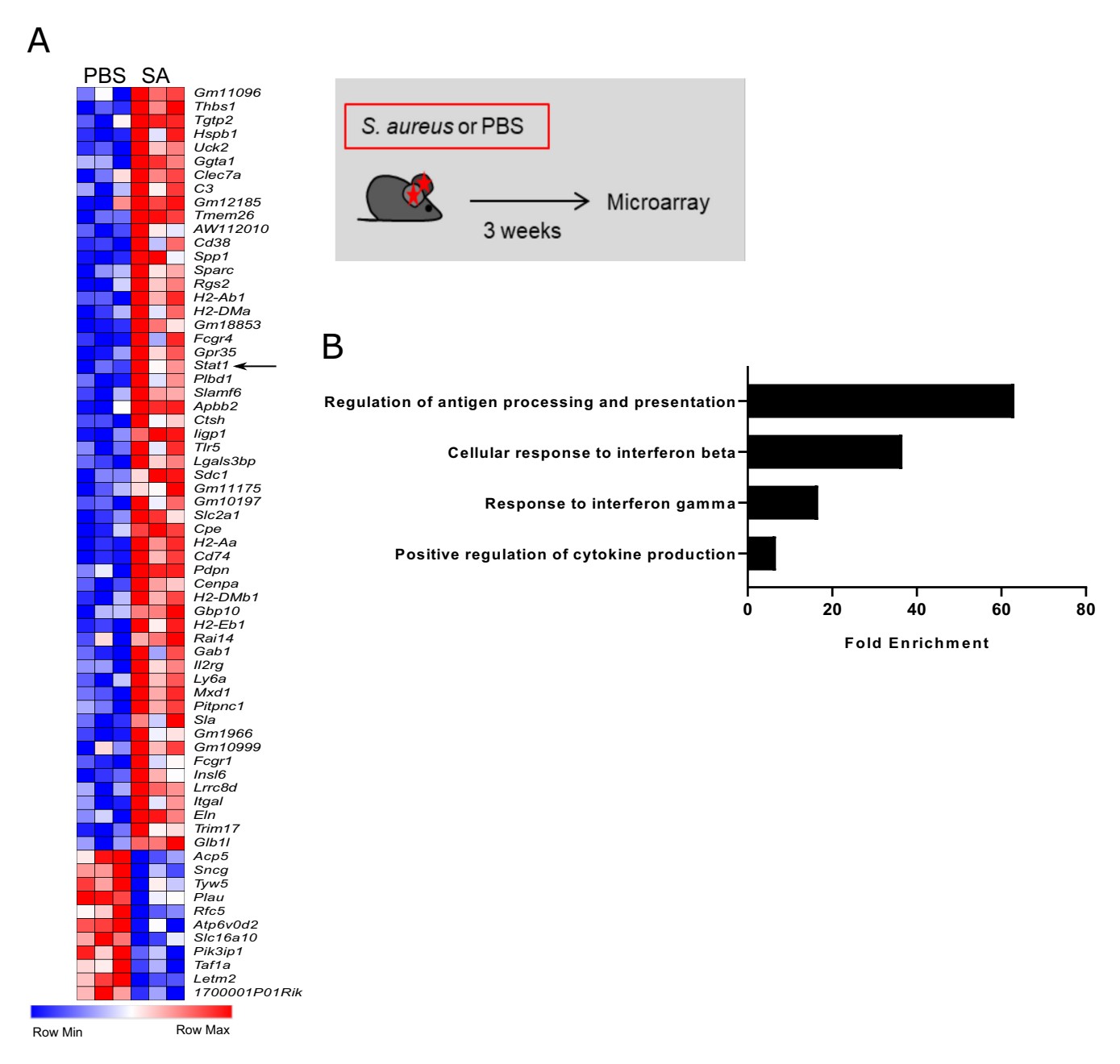

**Figure 4.** Innate memory signature in *S. aureus* challenged dermal Mφ. (A) Heat map depicting differentially expressed genes between PBS and *S. aureus* treated *Ccr2*[-/-] mice 3 weeks after infection. Total CD64[hi] macrophages were sorted from ear skin as described in *Figure 1—figure supplement 1A*. (B) Gene ontology analysis of overrepresented terms in PBS vs *S. aureus* infected mice. (Data represents three biological replicates per condition in one experiment). Genes with a fold change above 2.0 and a student's t test P value lower than 0.05 were analyzed using the Panther bioinformatics platform.

Ly6C[hi] cells invading the infected dermal microenvironment were primed as well. As Ly6C is downregulated by macrophages upon differentiation (*Kolter et al., 2019*), we sorted dermal Mφ from *Cx3cr1*[gfp/+] reporter mice 3 weeks after primary infection (without secondary infection). We sorted two populations, namely CX₃CR1[lo] Ly6C[lo] resident Mφ and CX₃CR1[int] Mφ which are partly Ly6C[hi] and derived from incoming monocytes. We found both subsets to exhibit increased expression of *Stat1* and *Cxcl9* compared to cells sorted from naïve mice. Interestingly, the expression of both genes was higher in the Ly6C[hi] subset, indicating that, while innate memory occurs in the absence of

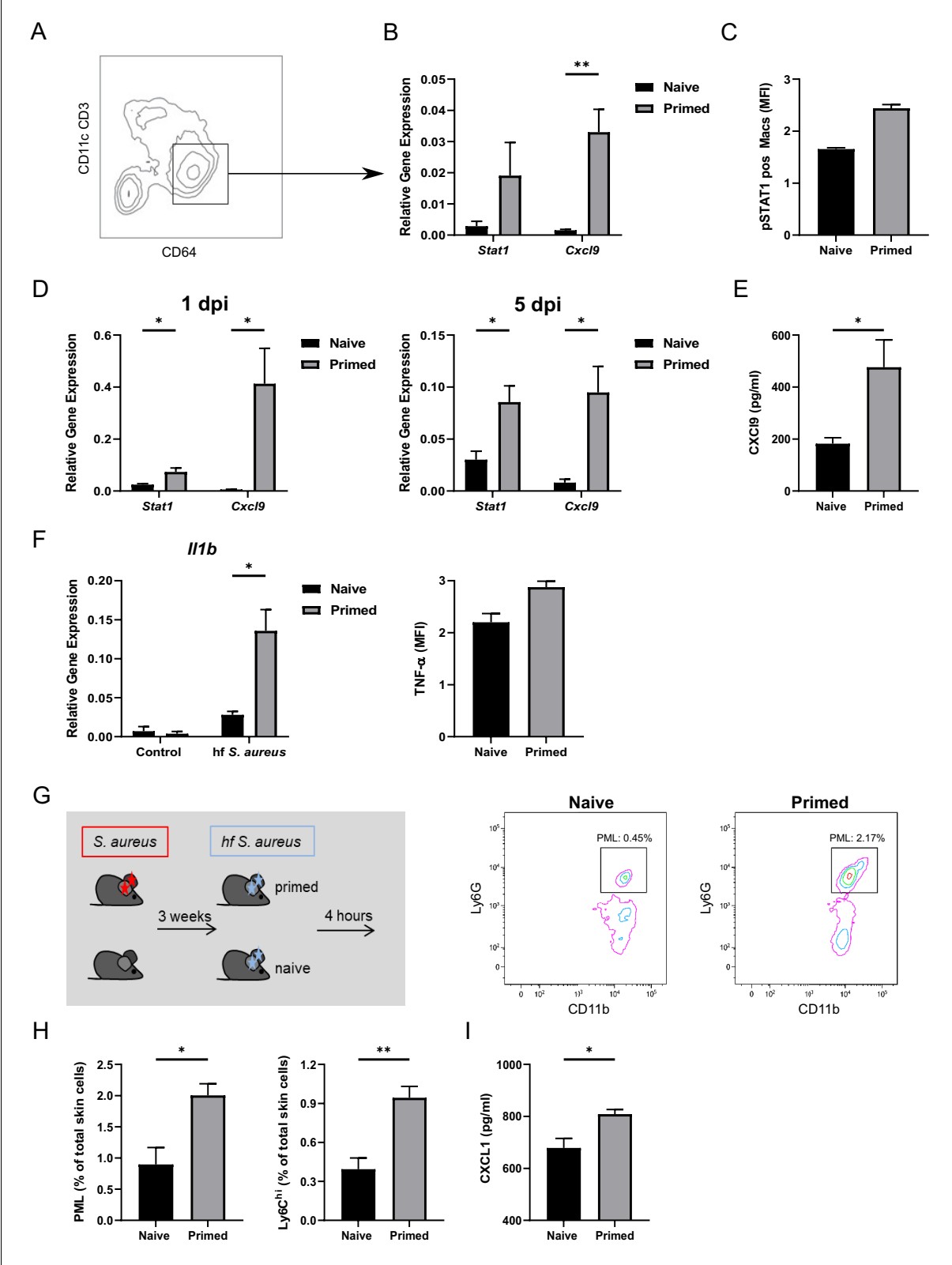

**Figure 5.** Dermal Mφ show a specific memory signature and increased responsiveness after (re-)infection ex vivo. (**A**) Exemplary FACS plot of CD64[hi] dermal Mφ from wt mice 3 weeks after primary infection. (**B**) RNA expression of all CD64[hi] dermal Mφ (as in A) sorted from *S. aureus* infected wt mice (primed) or naïve mice, analyzed 3 weeks after infection (n: five biological replicates from two independent experiments). (**C**) Digested ear skin from naïve or infected (primed) mice was analyzed for phosphorylated STAT1 positive dermal Mφ by flow cytometry (n: two biological replicates from two

*Figure 5 continued on next page*

Figure 5 continued

independent experiments). (**D**) RNA expression of sorted dermal Mφ from primed or naïve wt mice following (re-)infection periods of 1 and 5 days (n: three biological replicates from one independent experiment). (**E**) CXCL9 ELISA of skin lysates of primed or naïve wt mice after 5 days of (re-)infection (n: eight biological replicates representing two independent experiments). (**F**) Dermal Mφ sorted either from *S. aureus* infected mice (primed) or naïve wt mice, were re-challenged ex vivo with fixed *S. aureus*. Expression of pro-IL-1β was determined by qPCR. Intracellular (i.c) TNF concentrations were determined by antibody staining and analysis by flow cytometry. (**G–H**) Wt mice were infected i.d. with *S. aureus* ($10^7$ CFU in 10 μl PBS) and (re-)infected i.d. 3 weeks later (primed) or for the first time (naïve) with fixed bacteria ($10^8$ CFU in 10 μl PBS) and analyzed for skin infiltrating PML and Ly6C$^{hi}$ dermal Mφ (n: eight biological replicates from two independent experiments). (**I**) CXCL1 ELISA of skin lysates from mice treated as in G. Data were analyzed using either two tailed unpaired T test (B, D and F) or with two tailed Mann-Whitney test (E, H and I). Error bars are mean ± SEM. *p<0.05, **p<0.01,. The online version of this article includes the following figure supplement(s) for figure 5:

Figure supplement 1. CXCL9 inhibits the growth of *S. aureus* Newman.
Figure supplement 2. Innate memory signature is lost in clodronate treated mice.
Figure supplement 3. Ly6C$^{hi}$ dermal Mφ can be primed within the infected tissue.

incoming Ly6C$^{hi}$ Mφ, these cells can also be primed in the microenvironment of the infected ear (*Figure 5—figure supplement 3*).

Surprisingly, we did not find *S. aureus* to prime Mφ from different sources (bone-marrow, peritoneal cavity) for an enhanced secondary response in vitro as established for other effectors (*Schrum et al., 2018*; *Cheng et al., 2014*; *Ifrim et al., 2014*). Stimulation of bone marrow-derived Mφ with fixed *S. aureus* followed by re-stimulation 24 hr later did not enhance cytokine production. Likewise, priming of peritoneal Mφ for 24 hr with fixed *S. aureus* followed by a re-stimulation 5 days later failed to enhance intracellular TNF production (data not shown). This indicated that the induction of innate memory in vivo requires distinct cues from the infected tissue, although monocytes and adaptive immune cells are not involved (see above).

## Innate memory is a transient effect and partially independent of classical Toll-like receptor signaling and skin microbiota

MyD88 serves as a common adapter for Toll-like receptors (TLR), which are putative sensors for *S. aureus* and other skin commensals in dermal Mφ (*Feuerstein et al., 2015*). In order to establish the role of Myd88 in the innate memory model, we infected *Myd88$^{-/-}$* mice for 3 weeks and analyzed transcriptional priming (without secondary infection), as well as killing capacity and PML recruitment 5 days after re-challenge. Strikingly, primed *Myd88$^{-/-}$* mice had reduced bacterial burden and increased expression of *Stat1* and *Cxcl9* compared to naïve mice (*Figure 6—figure supplement 1A and B*). Furthermore, dermal Mφ from primed mice exhibited a robust *pro-IL1β* response after *S. aureus* ex vivo stimulation (*Figure 6—figure supplement 1C*). The above results indicate that innate *S. aureus* memory seems to be, at least partially, independent of MyD88.

Next we assessed the role of skin flora in macrophage priming. All mice tested so far were bred and kept under specific pathogen free conditions. These conditions did not exclude *S. aureus* colonization. We therefore tested our mouse strains for *S. aureus* colonization on a random basis and found *S. aureus* colonization in about one third of all tested mice (data not shown). This raised the question whether *S. aureus* colonization alone would be able to induce innate memory. We therefore monocolonized a cohort of sDMDMm2 mice (*Uchimura et al., 2016*; *Brugiroux et al., 2016*) – harboring a defined physiological intestinal microbiota but devoid of *S. aureus* – under axenic conditions with *S. aureus* by smearing a bacterial suspension onto their ear skin. Successful colonization was confirmed one and two weeks after initial ear coating. After three weeks, mice were infected intradermally with fixed bacteria for 4 hr. Skin colonization with *S. aureus* led to a 50% increase in the frequency of Ly6C$^{hi}$ Mφ after infection of sDMDMm2 mice while it did not affect the abundance of infection-induced PML (*Figure 6A*). *S. aureus*-infected SPF mice generally revealed higher numbers of PML and Ly6C$^{hi}$ Mφ compared to infected sDMDMm2 mice with or without *S. aureus* colonization. Importantly, the increase in both Ly-6C$^{hi}$ Mφ and PML induced by infection (SPF mice) was much greater as compared to the innate memory effect induced by skin colonization (sDMDMm2). Thus, *S. aureus* colonization slightly increases the response to subsequent infection, but does not induce an immunological memory similar to a dermal infection with the same organism.

A key feature of the adaptive immune system is the establishment of long-term memory, which protects the host from similar infections for months to years. In contrast, we hypothesized that innate

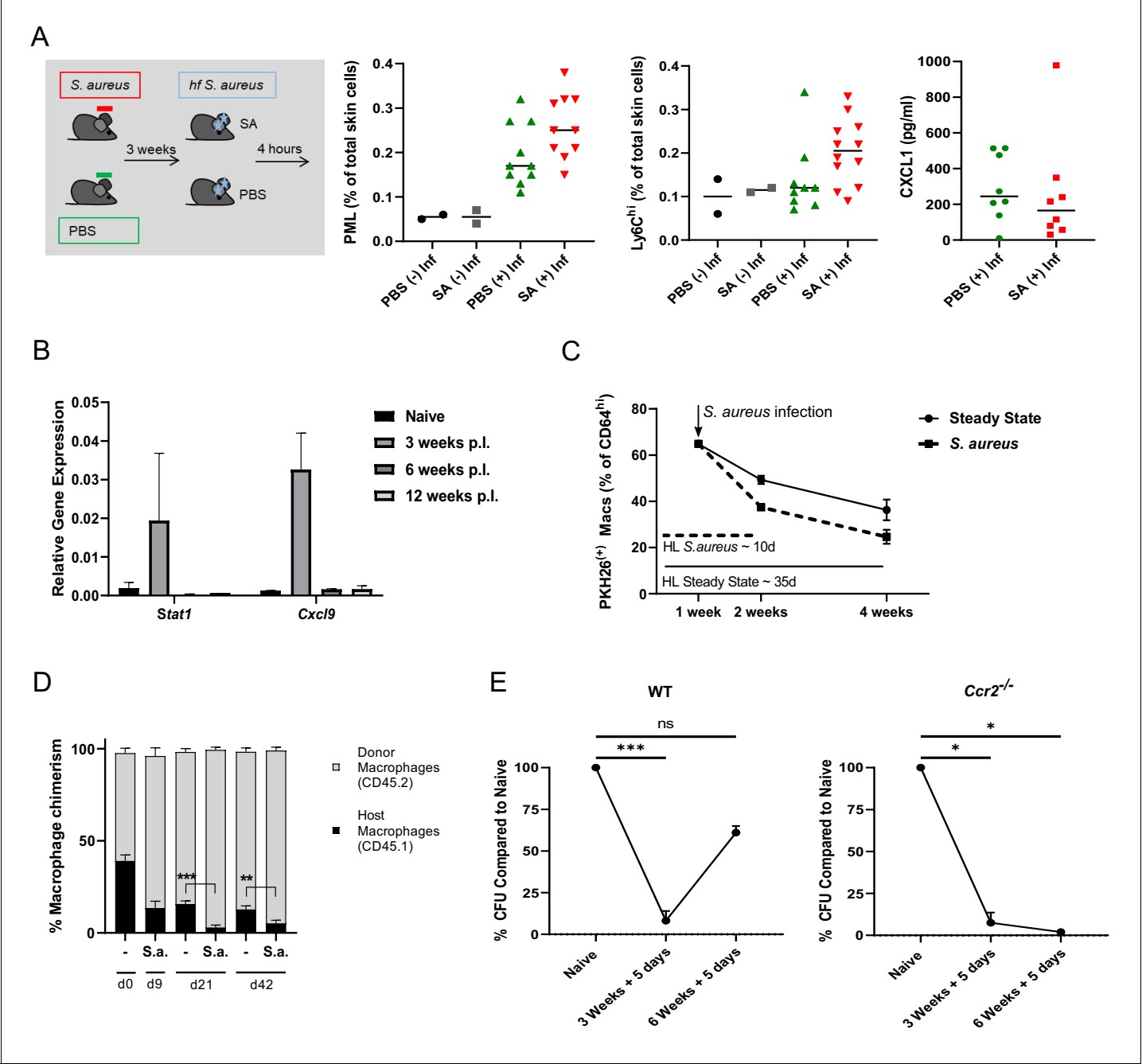

**Figure 6.** Innate memory is a transient effect and independent of skin microbiota. (**A**) C57BL/6J mice harbouring the sDMDMm2 microbiota were colonized with *S. aureus* under axenic conditions, inoculated 3 weeks later with fixed bacteria ($10^8$ CFU in 10 µl PBS) or PBS and analyzed for skin infiltrating PML, Ly6C$^{hi}$ monocytes and skin CXCL1 levels after 4 hr (n: 10–12 biological replicates representing two independent experiments). (**B**) RNA expression of sorted CD64$^{hi}$ dermal Mφ from *S. aureus* infected mice or naïve mice analyzed 3, 6 and 12 weeks after infection (n:2–4 biological replicates from two independent experiments). (**C**) Wt mice were treated i.v. with PKH26 red fluorescent cell linker. One week after labeling, mice were infected i.d. with *S. aureus*. After another 1 and 3 weeks, the contribution of PHK26 positive to all CD64$^{hi}$ dermal Mφ was quantified (n: two biological replicates per time point from two independent experiments). (**D**) B6-CD45.1 mice were transplanted with $10^7$ bone marrow cells from B6-CD45.2 congenic mice after fractional lethal irradiation. 8 weeks after transplantation dermal Mφ chimerism was determined by flow cytometry analysis using CD45.1 and CD45.2 antibodies (d0) or infected with *S. aureus* ($10^7$ CFU in 10 µl PBS) intradermally in the left ear. At the indicated time points the mice were sacrificed and dermal Mφ chimerism was determined by flow cytometry in the infected (S.a.) and uninfected ears (-). (n: three biological replicates for d0, and four biological replicates for all other groups). (**E**) Wt and *Ccr2*$^{-/-}$ mice were i.d. infected with *S. aureus* ($10^7$ CFU in 10 µl PBSl) and i.d. (re-) infected 3 or 6 weeks later with the same amount of bacteria and analyzed for bacterial load after an additional 5 days (n: three biological replicates

*Figure 6 continued on next page*

*Figure 6 continued*

from one independent experiment for the 6 week time points). Data were analyzed using two tailed unpaired T test. Error bars are mean ± SEM. *p<0.05, **p<0.01, ***p<0.001.

The online version of this article includes the following figure supplement(s) for figure 6:

**Figure supplement 1.** Innate memory is partially Myd88 independent.

**Figure supplement 2.** PKH26 labels Ly6C$^{lo}$ but not Ly6C$^{hi}$ dermal Mφ.

memory is intrinsic to the single cell and transient, that is it is non-clonal and not passed on to the next cell generation and lost, when the single trained cell is replaced. To test this, we analyzed dermal Mφ from naïve and primed mice after 6 and 12 weeks as well as the innate memory status of mice which were (re-)infected after 6 weeks. Already after 6 weeks, the innate memory signature, that is the increased expression of *Stat1* and its target gene *Cxcl9* were lost (*Figure 6B*). Furthermore, increased bacterial killing was lost in wt mice 6 weeks after infection (*Figure 6E*, left panel). To further investigate this phenotype, wt mice were intravenously treated with the PKH26 red fluorescent cell linker, which labels resident dermal Mφ for up to four weeks in vivo, whereas Ly6C$^{hi}$ incoming monocytes remain unlabeled (*Figure 6—figure supplement 2*). One week after labeling, mice were i.d. infected with *S. aureus* and one and three weeks after infection, PHK26 positive dermal Mφ were analyzed (*Figure 6C*). Compared to uninfected mice, *S. aureus* infection decreased the half-life of dermal Mφ from an estimated 35 days to 10 days, that is by around 70%. The increased turnover of dermal Mφ was confirmed by a complementary approach employing chimeras harboring bone marrow from congenic mice (B6-CD45.2 bone marrow in B6-CD45.1 mice), which allowed to discriminate previously resident Mφ from Mφ derived from the (transplanted) bone-marrow, combined with *S. aureus* infection. Eight weeks after transplantation of 10$^7$ wt bone marrow cells, mice were infected i.d. with *S. aureus*. Compared to uninfected mice, an increased replacement of the resident dermal Mφ pool by donor derived cells was observed up to six weeks after *S. aureus* infection (*Figure 6D*). We speculated that immune memory would be prolonged in *Ccr2$^{-/-}$* mice, in which monocyte input after infection is largely absent (*Figure 2A*). Indeed, increased bacterial killing could be observed in *Ccr2$^{-/-}$* mice following secondary infection in mice that had been primed 6 weeks previously (*Figure 6E*, right panel), while no difference in PML numbers was observed between primed and naïve mice (data not shown).

Taken together, these data indicate that *S. aureus*-induced innate memory is a transient phenomenon, which is lost within 6 to 12 weeks due the replacement of resident Mφ by monocyte-derived Mφ (model depicted in *Figure 7*).

## Discussion

Mucocutaneous resistance against potentially pathogenic bacterial colonizers largely relies on cellular innate immunity. It seems inherently plausible that evolution has equipped this system with means to adapt to repeated microbial encounters, since they are likely to reoccur. Accordingly, it has been appreciated for over 50 years that Mφ downmodulate the response to endotoxin after exposure, a phenomenon called tolerance (*Greisman and Hornick, 1975*). More recently, the opposite, for example propagation of cytokine transcription by a second microbial challenge, as compared to the first, has been linked to epigenetic changes and metabolic programming. Notably, much of the groundwork was done in vitro, and microbial components like β-glucan and LPS, and microorganisms, such as yeast, vaccine strain mycobacteria and herpes viruses have been found to reprogram Mφ (*Quintin et al., 2012*; *Yoshida et al., 2015*; *Kleinnijenhuis et al., 2012*; *Saeed et al., 2014*; *Barton et al., 2007*). Very recently *S. aureus* has also been shown to induce a Mφ dependent protective immunity in a penetrating skin and skin structure infection (*Chan et al., 2018*).

Systemic infections including those with focal onset, which are typically conceived by the host as sickness, and diffusible effectors are conceptually challenging, since they induce a cascade of multicellular intermediate events, including cytokines and soluble mediators. These, in turn, have the potential to change the activation state of distant immune cells and organs. In contrast, most challenges, which are imposed on the mucocutaneous innate immune system, include very local ground work, for example removal of bacteria or confinement to small abscesses, efferocytosis, tissue repair

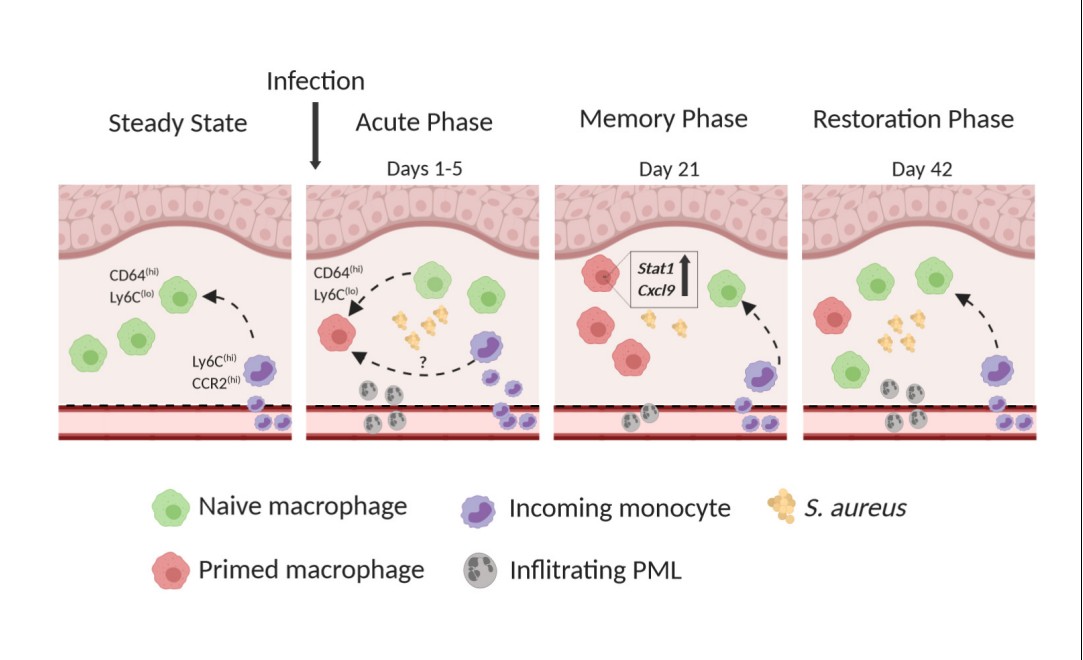

**Figure 7.** Model of innate immune memory during *S. aureus* skin infection. **Steady State**: CD64[hi], Ly6C[lo] Mφ have a half-life of 5–6 weeks and are replaced by CCR2[hi], Ly6C[hi] incoming monocytes. Within the tissue, monocytes downregulate CCR2 and Ly6C and upregulate CD64. **Acute Phase:** Resident CD64[hi] Mφ respond to primary *S. aureus* infection and adopt a memory phenotype, thus becoming primed for subsequent infection. However, infection results in reduced Mφ half-life and increased influx of CCR2[hi], Ly6C[hi] monocytes which may also undergo priming within the microenvironment. The acute phase is further characterized by high numbers of PMLs infiltrating the tissue. **Memory Phase**: *S. aureus* is cleared faster during secondary infection due to primed Mφ which are characterized by increased expression of *Stat1* and *Cxcl9*. Reduced numbers of skin infiltrating PMLs are present in the tissue compared to naïve mice. **Restoration Phase**: Innate immune memory is lost due to continual influx of Ly6C[hi] monocytes which give rise to naïve Mφ and replace primed Mφ. Naïve Mφ lack increased expression of *Stat1* and *Cxcl9* resulting in slower bacterial clearance and increased skin PMLs after secondary infection.

etc. They do not involve activation of systemic immunity. Here, we aimed to mimic this daily grind of dermal Mφ with a discrete and strictly intradermal staphylococcal infection model, in order to better grip mechanisms underlying cell autonomous priming by bacteria residing in immediate proximity. This model recapitulates the initiation of the immune response by resident dermal Mφ, the recruitment of PML and monocytes to the site of infection, the killing of bacteria and the contraction of inflammation without measurable circulation of bacteria and cytokines, and without effect on the mouse wellbeing (*Feuerstein et al., 2017*; *Feuerstein et al., 2015*). Thus, it differs sharply from subcutaneous *S. aureus* infections, which lead to strong inflammation, large abscesses, dermonecrosis and bacterial dissemination to organs like the spleen, kidney and liver (*Chan et al., 2018*). Another caveat of severe infection models with bacterial dissemination, for example to the bone-marrow, is the difficulty in excluding long term survival of bacteria in body niches. In the case of *S. aureus*, bacteria have been found to survive in osteoblasts from mice for months (*Tuchscherr et al., 2015*).

We found here that mice showed stronger PML and monocyte recruitment and bacterial killing early in dermal staphylococcal infection, if it had been preceded by an intradermal staphylococcal infection three weeks before, indicating that lasting immune priming indeed occurred at the local level. The memory effect was strictly dependent on skin resident dermal Mφ, and did not require mature T, B and NK cells, as well as monocytes. The transcriptional effects of dermal Mφ memory in situ were rather subtle, since intradermal *S. aureus* infection led to persistent transcriptional changes in only 65 genes compared to vehicle control. Yet, the circumscribed transcriptional changes acquired in vivo, which translated into an altered inflammatory response to *S. aureus* ex vivo, further substantiated the interpretation that Mφ are primed for an enhanced secondary response on the single cell level.

It seems notable that the *S. aureus*-induced memory signature in dermal Mφ comprised activation of STAT1, whereas type I inflammatory genes were not increased. It appears that the immediate

innate immune response is fully resolved 2 weeks after infection in our model, as indicated by the absence of culturable bacteria, of skin infiltrating PML and of type I cytokines in the tissue at this stage. Among the signaling intermediates, metabolic and epigenetic changes which have been previously implicated in Mφ memory (*Quintin et al., 2012*; *Yoshida et al., 2015*; *Cheng et al., 2014*; *Saeed et al., 2014*) STAT1 appears to play a dominant role (*Yoshida et al., 2015*; *Saeed et al., 2014*). It is intriguing that the STAT1-dependent chemokine CXCL9 possess direct antimicrobial activity independent of its chemotactic activity (*Reid-Yu et al., 2015*; *Yang et al., 2003*). However, whether this CXCL9 property directly impacts on the defense against *S. aureus* in the dermis remains to be established. A major observation was that substantial memory lasted for less than six weeks in vivo. This finding is in line with data from Garcia-Valtanen et al., who observed a decline in the β-glucan induced memory in vivo in a similar time span (*Garcia-Valtanen et al., 2017*). Several lines of evidence suggest that 'naive' bone marrow-derived cells dilute out the resident Mφ population and thereby terminate immunological memory in local *S. aureus* infection. First, monocyte-deficient mice exhibited somewhat longer lasting immunological memory as compared to wt mice. Next, we did not find *S. aureus* to convincingly prime monocytes, bone marrow-derived Mφ or peritoneal Mφ in vitro. In other words, innate memory appears to require terminally differentiated Mφ in the tissue environment. This is in contrast to what has been described for soluble TLR ligands and other microorganisms (*Schrum et al., 2018*; *Cheng et al., 2014*; *Ifrim et al., 2014*). Moreover, the memory phenotype could not be transferred by bone-marrow transplantation. However, the transfer experiments showed that *S. aureus* infection led to a strong redistribution of the skin myeloid cell subsets with high numbers of infiltrating bone marrow-derived monocytes. This indicated a strong dilution of resident dermal Mφ within 6 weeks. Of note, recipient mice were irradiated without shielding of the heads, a method to protect resident cells of the ear skin from irradiation damage. This could hinder proliferation of resident dermal Mφ after transplantation and after *S. aureus* infection, when it is normally high (own unpublished data) and could putatively maintain innate memory conditions. We therefore simulated *S. aureus*-induced changes of the dermal Mφ compartment with an additional in vivo model where we labelled specifically resident dermal Mφ, and not bone marrow-derived monocytes (own observations), with the phagocytic cell dye PKH26. We found that the normal half-life of dermal Mφ (35 days in homeostasis) was shortened to approximately 10 days after *S. aureus* infection.

The skin constitutes a remarkable host-microbe interface, where micrometers decide between harmless colonization by *S. aureus*, which occurs in up to 30% of humans (*Noskin et al., 2007*) and invasive infection. However, the physical barrier of the epidermis appears to be less tight than commonly thought, since nucleic acids from *S. aureus* can be found in the non-lesional dermis of up to 30% of people (*Crisp et al., 2015*). Accordingly, we wondered, whether *S. aureus*, as a regular component of the skin microbiota, would subject dermal Mφ to constant innate memory, thereby compensating for rapid replacement of programmed *M*φ. Analysis of transiently *S. aureus* colonized sDMDMm2 mice hinted at innate memory effects by colonization, yet the effects were modest. However, it remains an attractive hypothesis that *S. aureus* gets access to the dermis via disruption of the epidermis by minimal mechanical trauma and through preformed gaps, for example hair follicles. This may propagate the dynamic and flexible nature of the dermis as a large immune organ with high adaptation to colonizing bacteria.

In summary, tissue resident dermal Mφ are programmed independently of bone marrow-derived monocytes during staphylococcal skin infection to mediate increased resistance against a second infection. However, the infection-induced turnover of resident skin Mφ limits the duration of this innate memory response.

We propose that reprogramming of Mφ serves different purposes related to pathogen-specific infection biology. In the case of colonizing or minimally invasive *S. aureus*, innate memory is restricted to the dermal niche and wanes over weeks. This local recall phenomenon is reminiscent of the recently reported priming of alveolar Mφ in local adenovirus infections (*Yao et al., 2018*). On the other hand, systemic spread of staphylococci like in sepsis is relatively rare and carries a high risk of death. Thus, under these conditions, innate memory is likely futile. In contrast, microorganisms with an intracellular lifestyle and limited virulence, for example mycobacteria, may induce innate memory of cells in central components like hematopoietic stem cells in order to allow for individual adaptations in host resistance (*Kaufmann et al., 2018*).

# Materials and methods

## Key resources table

| Reagent type (species) or resource | Designation | Source or reference | Identifiers | Additional information |
|---|---|---|---|---|
| Genetic reagent (*Mus. musculus*) | C57BL/6(J) | Jackson lab | Stock No: 000664 | |
| Genetic reagent (*Mus. musculus*) | C57BL/6(N) | Jackson lab | Stock No: 005304 | |
| Genetic reagent (*Mus. musculus*) | *Ccr2-/-* | Gift of Marco Prinz, (University of Freiburg) | | |
| Genetic reagent (*Mus. musculus*) | $Rag2^{-/-}\gamma c^{-/-}$ | Gift of Tilman Brummer (University of Freiburg) | | BALB/c background |
| Genetic reagent (*Mus. musculus*) | $Cx3cr1^{gfp/+}$ | Gift of Steffen Jung (Weizmann Institute of Science) | | |
| Genetic reagent (*Mus. musculus*) | *Myd88-/-* | **Deshmukh et al., 2012** | | |
| Genetic reagent (*Mus. musculus*) | sDMDMm2 | Clean Mouse Facility, University of Bern, Switzerland | | C57BL/6(J) background |
| Antibody | anti-mouse CD45 eFluor450 | Invitrogen | Cat: 48-0451-82 Clone: 30-F11 | 1:300 |
| Antibody | anti-mouse CD11b PE-Cy7 | Invitrogen | Cat: 25-0112-82 Clone: M1/70 | 1:3000 |
| Antibody | anti-mouse Ly6G FITC | BD Pharmingen | Cat: 551460 Clone: 1A8 | 1:200 |
| Antibody | anti-mouse Ly6C PerCP-Cy5.5 | BD Pharmingen | Cat: 560525 Clone: AL-21 | 1:500 |
| Antibody | anti-mouse CD64 PerCP/Cy5.5 | Biolegend | Cat: 139301 Clone: X54-5/7.1 | 1:100 |
| Antibody | anti-mouse CD11c APC | BD Pharmingen | Cat:561119 Clone HL3 | 1:100 |
| Antibody | mouse MHC class II eFluor450 | eBioscience | Cat: 48-5320-82 Clone: AF6120.1 | 1:400 |
| Antibody | anti-mouseCD3e | Invitrogen | Cat: 12-0031-82 Clone: 145–2 C11 | 1:200 |
| Antibody | anti-mouse Ly6C PE | Invitrogen | Cat: 12-5932-80 Clone: HK1.4 | 1:1000 |
| Antibody | Anti-mouse pSTAT1 AF647 | BD Biosciences | Cat:612597 Clone: 4a | |
| Chemical compound/drug | liposomal clodronate | Clodrosome Encapsula NanoSciences | Cat: CLD-8909 | |
| Chemical compound/drug | PKH26 Red | Sigma Aldrich | Cat: PKH26GL | |
| Software | GraphPad Prism | https://www.graphpad.com/scientific-software/prism/ | | Version 8 |
| Software | Kaluza analysis | https://www.beckman.de/flow-cytometry/software/kaluza | | |

## Mice

All mice were on C57BL/6J, C57BL/6N or Balb/c genetic background. Mice lacking MyD88 were previously described (**Deshmukh et al., 2012**). Mice lacking CCR2 were kindly provided by Marco Prinz (University of Freiburg). $Rag2^{-/-}\gamma c^{-/-}$ mice were kindly provided by Tilman Brummer (University of Freiburg). $Cx3cr1^{gfp/+}$ mice were a kind gift of Steffen Jung (Rehovot, 76100, Israel). C57BL/6(J)

mice carrying the stable defined moderately diverse mouse microbiota 2 (sDMDMm2) (*Uchimura et al., 2016*; *Brugiroux et al., 2016*) were born and maintained in flexible-film isolators in the Clean Mouse Facility, University of Bern, Switzerland. Age and gender-matched mice were used at 8–9 weeks.

### Bacterial strains

*S. aureus* strain Newman was used. Bacteria were grown in Luria-Bertani broth medium to exponential growth phase, washed with PBS and resuspended in Dulbecco's Phoshate Buffered Saline (PBS). Bacterial concentrations were determined using an optical spectrophotometer (600 nm, $OD_{600}$). Colony forming units (CFUs) of the inoculum were verified by dilution series plated onto blood agar plates (COS, Biomerieux) overnight at 37°C. In some experiments, bacteria were heat-fixed (80°C, 30 min) after adjusting the cell number to $10^9$ CFUs/ml as previously described (*Feuerstein et al., 2015*). For colonization studies *S. aureus* suspension ($10^9$ CFUs/ml) was smeared onto both ears of the mice under axenic conditions using sterile cotton swab. Ears of control mice were smeared with a sterile PBS solution. Successful colonization of mice was tested by swabbing the ears of mice with a PBS-soaked cotton swab and streaking directly onto mannite sodium chloride plates.

### Mouse model of intradermal skin (re)-infection

All procedures were approved by the Regional Commission of Freiburg, Baden - Württemberg. Approximately $10^7$ colony forming units (CFUs) *S. aureus* in 10 μl PBS were intradermally (i.d.) injected (30 gauge needle, U-100 insulin syringe (BD)) into the ear pinna of anaesthetized mice. In some experiments, heat fixed (hf) bacteria were used for i.d. inoculation ($10^8$ hf bacteria in 10 μl PBS per ear pinna). For re-infection experiments either $10^7$ CFU *S. aureus* in 10 μl PBS or $10^8$ hf bacteria in 10 μl PBS were i.d. injected into the same ear pinna 3 weeks (if not indicated otherwise) after the first infection. If not indicated otherwise groups of 4–5 mice were used per experiment followed, by at least one repetition to confirm the results. Lesion morphology was documented by digital photographs (Canon PowerShot A650 IS) of mice ears and analyzed by the software program Adobe Photoshop CS6.

### Depletion of resident skin M$\varphi$ in vivo

Resident skin Mφ were depleted by liposomal clodronate (Clodrosome Encapsula NanoSciences) as previously described (*Feuerstein et al., 2015*). Approximately, 10 μl of liposomal clodronate or control liposomes were injected i.d. into the ear pinna of anaesthetized mice. *S. aureus* was injected i.d. into the same ear pinna 10d later.

Labeling of dermal macrophages with PKH26 Red Fluorescent Cell Linker wt mice were intravenously injected with 100 μl PKH26/Diluent B mix (0.1 mM PKH26 in 100 μl Diluent B). Labeling efficiency of resident dermal Mφ was analyzed by FACS and stable up to 4 weeks (data not shown). 1 week after labeling mice were infected with $10^7$ CFU *S. aureus* i.d. into both ears and labeling or rather turnover of resident Mφ followed and analyzed for additional 3 weeks by FACS.

### Mouse bone marrow chimera

B6-CD45.1-mice (Ly5.1) were fractionally irradiated with $2 \times 4.5$ Gy and reconstituted with $10^7$ nucleated bone marrow cells from congenic B6-CD45.2 mice. After 8 weeks a > 99% donor chimerism for blood monocytes was confirmed by flow cytometry analysis using CD45.1 and CD45.2 antibodies. At this timepoint, untreated mice were sacrificed and chimerism of dermal macrophages was analyzed using CD45.1 and CD45.2 antibodies (d0). The other groups of mice were infected with $10^7$ CFU *S. aureus* in 10 μl PBS intradermally into the left ear (as described above). Treated mice were sacrificed at the indicated time points and dermal Mφ were isolated and analyzed by flow cytometry as described above to determine Mφ chimerism in the infected ear (S.a.) and the uninfected ear (-).

### Mouse bone marrow chimera with infected bone marrow

Ly5.1 (CD45.2) mice were i.d. infected with $10^7$ CFU *S. aureus* in 10 μl PBS. 3 weeks later wt (CD45.1) mice were fractionally irradiated with $2 \times 4.5$ Gy and reconstituted with $10^7$ nucleated bone marrow cells from the infected (CD45.2) mice. After 2 weeks irradiated wt mice were infected with $10^7$ CFU *S. aureus* i.d. into the ear for 5 days (as described above).

## Immune cell phenotyping of skin tissue

Mouse ears were subjected to enzymatic digestion as previously described (*Feuerstein et al., 2015*). After digestion, samples were filtered with a 70 µm cell strainer (BD), washed with FACS Buffer (PBS+2%FBS+2 mM EDTA) and stained with the indicated antibodies. The following antibodies were used: anti-mouse CD45 eFluor450 (eBioscience), anti-mouse CD11b PE-Cy7 (eBioscience), anti-mouse Ly6G FITC (BD Biosciences), anti-mouse Ly6C PerCP-Cy5.5 (BD Biosciences), anti-mouse CD64 PerCP/Cy5.5 (Biolegend), anti-mouse CD11c APC (BD Biosciences), mouse MHC class II eFluor450 (eBiosciences) and anti-mouse CD3e (Biolegend). Cell samples were analyzed with a 10-color flow cytometer (Gallios, Beckman Coulter) and the Kaluza software (version 1.5a, Beckman Coulter).

## Intracellular phosphorylated Stat1 staining

Mouse ears were subjected to enzymatic digestion, filtered, washed and cells finally resuspended in 100 µl RPMI medium supplemented with 10% FBS and antibiotics (ciprofloxacin, 10 mg/ml). Then cells were stimulated for 15 min at 37°C with 200 ng/ml IFN$\gamma$ and subsequently fixed with 100 µl fixation buffer (Cytofix Fixation Buffer, BD Biosciences) for 10 min at 37°C. After centrifugation and permeabilization with 400 µl ice cold Perm Buffer III (BD Phosflow) for 30 min at 4 °C cells were washed twice with PBS containing 1% BSA. Then cell surfaces were stained with anti-mouse CD45 eFluor450 (eBioscience), anti-mouse CD11b PE-Cy7 (eBioscience), anti-mouse CD11c APC (BD Biosciences) and intracellular phosphorylated Stat1 stained with mouse anti-Stat1 (pY701) Alexa Fluor 647 (BD Biosciences) 30 min at RT and subsequently analyzed by flow cytometry.

## Tissue embedding and staining

Ear skin was collected, bisected and ear halves were embedded in Tissue-Tek O.C.T. compound (Sakura Finetek Europe B.V.) and subsequently frozen inliquid nitrogen. 8 µm cryosections were taken with a Leica CM 1850 (Leica Biosystems, Nussloch, Germany) and stained with H and E. Photographs were taken on a Zeiss Axioskop 50 (Zeiss, Jena, Germany) with the Olympus LC20 camera (Olympus Germany, Hamburg, Germany).

## Enzyme-linked immunosorbent assay (ELISA)

Ears were mechanically homogenised by Tissue Lyser, Qiagen. IL-1$\beta$, CXCL9 and KC concentrations in the homogenised skin tissue were quantified by ELISA according to the manufacturer's instructions (R and D Systems).

## Quantification of Colony Forming Units

Mice were sacrificed. Then ears were cut off at the hair line and homogenized in a Tissue Lyser, Qiagen. CFUs in the homogenized skin tissue were determined by serial dilutions on blood agar plates (COS, Biomerieux).

## CXCL9 killing assay

To determine CXCL9 antimicrobial activity, *S. aureus* was grown to $OD_{600}$ (0.4) and then incubated at 37°C, with shaking, for a further 2 hr in the presence of CXCL9 (4 µg/ml or 40 µg/ml) or PBS as control. Bacteria survival was determined by serial dilutions on blood agar plates.

## Ex vivo stimulation of dermal M$\varphi$, RNA Preparation and Reverse Transcriptase Polymerase Chain Reaction (qRT-PCR)

Mice skin was sterilized with 70% ethanol and subjected to enzymatic digestion by sterile skin digestion solution as per description. After 2 hr, 37°C and 1400 rpm, samples were filtered, washed with PBS and stained anti-mouse CD45 eFluor450 (eBioscience), anti-mouse CD11b PE-Cy7 (eBioscience), anti-mouse CD64 PerCP/Cy5.5 (Biolegend), anti-mouse CD11c APC (BD Biosciences) and anti-mouse CD3e (Biolegend). Dermal M$\varphi$ (CD45$^{hi}$ CD11b$^{hi}$ CD64$^{hi}$ CD3$^{lo}$ CD11c$^{lo}$) were sorted by FACS (MoFlow Astrios), resuspended in RPMI medium supplemented with 10% FBS with antibiotics (ciprofloxacin, 10 mg/ml) and plated in 48 well plates (30,000 cells/well, Corning Costar). The next day, adherent dermal M$\varphi$ were washed once and stimulated with medium containing hf *S. aureus* (multiplicity of infection (MOI) 50) for 2 hr at 37 °C C or with medium without hf *S.aureus* as control. Total

RNA was extracted using the RNAeasy micro kit, according to instruction manual (Qiagen). qRT-PCR was performed as previously described (*Deshmukh et al., 2012*). For expression levels without restimulation, dermal Mφ (CD45$^{hi}$ CD11b$^{hi}$ CD64$^{hi}$ CD3$^{lo}$ CD11c$^{lo}$) were sorted by FACS (MoFlow Astrios), and directly resuspended in RLT lysis buffer (Qiagen). Total RNA was extracted using the RNAeasy micro kit, according to instruction manual (Qiagen). The following mouse primer sequences were used (5′−3′): murine GAPDH fw ACTCCACTC ACGGCAAATTC, murine GAPDH rev TCTCCA TGGTGGTGAAGACA; murine pro IL-1β fw GTTGACGGACCCCAAAAGAT, murine pro IL-1β rev CCACGGAAAGACACAGGTA; murine Stat1 fw GGCGTCTATCCTGTGGTACAACA, murine Stat1 rev GTGACTGATGAAAACTGCCAACTC; murine Cxcl9 fw CGGACTTCACTCCAACACAG, murine Cxcl9 rev TAGGGTTCCTCGAACTCCAC. Unless otherwise stated, all CD64$^{hi}$ dermal Mφ were analyzed irrespective of Ly6C expression.

### GeneChip microarray assay

*Ccr2$^{-/-}$* mice were i.d. infected with *S. aureus* or PBS into both ear pinnae. Three weeks later dermal Mφ (CD45$^{hi}$ CD11b$^{hi}$ CD64$^{hi}$ CD3$^{lo}$ CD11c$^{lo}$) were sorted by FACS (MoFlow Astrios), and directly resuspended in RLT lysis buffer (Qiagen). Total RNA was extracted using the RNAeasy micro kit, according to instruction manual (Qiagen). Further sample processing was performed at an Affymetrix Service Provider and Core Facility, 'KFB - Center of Excellence for Fluorescent Bioanalytics' (Regensburg, Germany; www.kfb-regensburg.de). In brief, 1 ng of total RNA was used for reverse transcription to synthesize single-stranded (ss) cDNA with a T7 promoter sequence at the 5′ end. Next a 3′ adaptor was added and the ss cDNA was converted to double-stranded cDNA, which acted as a template for a pre- in vitro transcription (IVT) amplification by a nine cycle PCR. Afterwards antisense RNA (complimentary RNA or cRNA) was synthesized and linearly amplified by an in vitro transcription (IVT) of the double-stranded cDNA template. 20 μg of cRNA was purified and reverse transcribed into double-stranded (ds) cDNA, whereat unnatural dUTP residues were incorporated at a fixed ratio relative to dTTP. Purified ds cDNA was fragmented using a combination of uracil DNA glycosylase (UDG) and apurinic/apyrimidinic endonuclease 1 (APE 1) at the dUTP residues followed by a terminal labeling with biotin. 5,5 μg fragmented and labeled ds cDNA were hybridized to Affymetrix Clariom S mouse arrays for 16 hr at 45°C in a GeneChip hybridization oven 640. Hybridized arrays were washed and stained in an Affymetrix Fluidics Station FS450, and the fluorescent signals were measured with an Affymetrix GeneChip Scanner 3000 7G. Fluidics and scan functions were controlled by Affymetrix GeneChip Command Console v4.1.3 software.

### Microarray data analysis

Summarized probe set signals in log2 scale were calculated by using the GCCN-SST-RMA algorithm with the Affymetrix GeneChip Expression Console v1.4 Software. After exporting into Microsoft Excel, average signal values, comparison fold changes and significance P values were calculated. Probe sets with a fold change above 2.0 fold and a student's t test P value lower than 0.05 were considered as significantly regulated. GEO accession number: GSE145094.

## Acknowledgements

We thank Professor Andrew Macpherson, the Genaxen foundation and the Clean Mouse Facility of the University of Bern for their support to carry out gnotobiotic experiments. In addition, we are indebted to Anita Imm, the Lighthouse Core Facility and the Center for Experimental Models and Transgenic Servive (CEMT) of the University Medical Center Freiburg for their excellent technical support and assistance with the animal studies performed in this work.

## Additional information

### Funding

| Funder | Grant reference number | Author |
| --- | --- | --- |
| Deutsche Forschungsgemeinschaft | HE3127/9 | Philipp Henneke |

| Deutsche Forschungsgemeinschaft | HE3127/12 | Philipp Henneke |
| Deutsche Forschungsgemeinschaft | SFB/TRR167 | Philipp Henneke |
| Bundesministerium für Bildung und Forschung | 01EO0803 | Philipp Henneke |
| Bundesministerium für Bildung und Forschung | 01GL1746A | Philipp Henneke |
| Bundesministerium für Bildung und Forschung | 01EK1602A | Philipp Henneke |
| Else Kröner-Fresenius Foundation | | Philipp Henneke |
| H2020 Marie Skłodowska-Curie Actions | 744257 | Jakob Zimmermann |
| Schweizerischer Nationalfonds zur Förderung der Wissenschaftlichen Forschung | Ambizione PZOOP3_168012 | Mercedes Gomez de Agüero |

The funders had no role in study design, data collection and interpretation, or the decision to submit the work for publication.

### Author contributions

Reinhild Feuerstein, Conceptualization, Formal analysis, Investigation, Writing - original draft, Writing - review and editing; Aaron James Forde, Formal analysis, Investigation, Writing - original draft, Writing - review and editing; Florens Lohrmann, Julia Kolter, Formal analysis, Investigation, Writing - review and editing; Neftali Jose Ramirez, Jakob Zimmermann, Mercedes Gomez de Agüero, Investigation; Philipp Henneke, Conceptualization, Resources, Formal analysis, Supervision, Validation, Writing - original draft, Project administration

### Author ORCIDs

Aaron James Forde (iD) https://orcid.org/0000-0003-3754-5541
Philipp Henneke (iD) https://orcid.org/0000-0001-7314-7984

### Ethics

Animal experimentation: All animal experiments were approved by the Federal Ministry for Nature, Environment and Consumer's protection of the state of Baden-Wuerttemberg.

### Decision letter and Author response

Decision letter https://doi.org/10.7554/eLife.55602.sa1
Author response https://doi.org/10.7554/eLife.55602.sa2

## Additional files

### Supplementary files

• Transparent reporting form

### Data availability

All data generated or analysed during this study are included in the manuscript and supporting files. Source data files have been provided.

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
