## [Decision Letter]

**Acceptance summary:**

In this study the authors investigate the role of dermal macrophages in mediating innate immune memory during *S. aureus* infection. The authors use a model where intradermal priming in the ear with *S. aureus* increased bacterial killing upon reinfection, and propose that local innate memory may be involved, mediated by resident macrophages.

**Decision letter after peer review:**

Thank you for submitting your article "Resident macrophages acquire innate immune memory in staphylococcal skin infection" for consideration by *eLife*. Your article has been reviewed by three peer reviewers, one of whom is a member of our Board of Reviewing Editors, and the evaluation has been overseen by Satyajit Rath as the Senior Editor. The following individual involved in review of your submission has agreed to reveal their identity: James Cheng (Reviewer #3).

The reviewers have discussed the reviews with one another and the Reviewing Editor has drafted this decision to help you prepare a revised submission.

Summary:

The authors use a model where intradermal priming in the ear with *S. aureus* increased bacterial killing upon reinfection. The data suggests that the local innate memory may be involved, in particular mediated by resident macrophages. While priming altered the transcriptional signature of resident macrophages, such changes were transient, which was attributed to increased turnover of resident macrophages after priming.

While the issue of innate memory is of interest, many aspects of this study were hard to follow. Key controls were missing or not done correctly or experiments were not explained in enough detail, confounding interpretation of important experiments.

Essential revisions:

1) An important issue that does not seem to have been addressed adequately is the relative contributions of embryonic resident macrophages, inflammatory monocytes, and monocyte-derived resident macrophages to the innate memory. The authors should use an appropriate BM chimera (head protection) or another similar strategy to address this issue.

2) Many experiments were hard to follow (e.g., Figure 6D), and need to be explained in more detail.

3) Gating strategy needs to be shown (e.g., Figure 1C, D, Figure 4, Figure 5—figure supplement 2A).

4) Authors should be explicit (e.g., Ly6C^hi^ or Ly6C^lo^ macs) and consistent with macrophage nomenclature throughout study.

5) Authors should show an analysis of Ly6C^hi^ macs in Figure 1G.

6) Figure 3A: It seems like the authors treated primed mice with clodronate and naïve mice with control liposomes. In which case, this cannot be the corresponding FACS analysis for Figure 3B. If so, Figure 3A and B should be repeated by treating primed mice with clodronate or control liposomes, then showing effects on bacterial burden and with the corresponding FACS profiles to indicate depletion of dermal macrophages.

7) The authors need to show PKH26 labeling specificity.

8) The CFU is strikingly different between Figures 2B and 1C. wt is not included in Figure 2B, leading one to wonder what wt may have looked like. The authors should repeat to include wt in same experiment, or at least the authors need to explain this issue.

9) Regarding the half-life of resident dermal macrophages, Figure 1F seems to be pointing to more dermal Ly6C^lo^ macrophage 5 dpi while in Figure 6C, it seems *S. aureus* infection is decreasing the dermal macrophage number. Are the two sets of findings discrepant? Again, an analysis of Ly6C^hi^ macs in Figure 1G would help clarify the issue.

10) Figure 3A: absolute numbers need to be shown.

---

## [Author Response]

Essential revisions:1) An important issue that does not seem to have been addressed adequately is the relative contributions of embryonic resident macrophages, inflammatory monocytes, and monocyte-derived resident macrophages to the innate memory. The authors should use an appropriate BM chimera (head protection) or another similar strategy to address this issue.

We fully agree that the relative contribution of putatively distinct tissue mononuclear cells is at the heart of the study. We acknowledge that we did not fully succeed in communicating our substantial efforts to address this question. Our previous approach was manifold.

⦁ Inflammatory monocytes:

– First, we analyzed CCR2-deficient mice, where circulating monocytes (Feuerstein et al., 2015) and, as confirmed in Figure 2A and Figure 2—figure supplement 1A of the current manuscript, the monocyte progeny Ly6C^hi^ dermal macrophages (Kolter et al., 2019) are drastically reduced in staphylococcal skin infection. We found monocyte-deficient mice to develop “innate memory” similarly to wild type mice, i.e. CCR2-deficient mice showed enhanced bacterial killing and clearing of infection after a second staphylococcal challenge.

– Second, we showed that a distant dermal infection (one ear) does not induce memory in the other ear (Figure 3D), although both ears are connected via the blood stream and thus circulating monocytes. This suggests that memory is induced by local, resident cells.

– Third, we transplanted bone-marrow (the source of monocytes) from a challenged mice to a naïve mice, which did not induce memory (Figure 2—figure supplement 1B).

In summary, we did not find any evidence for monocytes being essential for innate immune memory in staphylococcal skin infection.

⦁ Embryonic resident macrophages:

The clodronate-depletion model strongly indicated that resident cells, and not newly arising bone-marrow derived macrophages provided memory function. Yet, dermal macrophages are a mixed population in relation to origin. The postnatal half-life of the macrophages is approximately 8 weeks, i.e. at the age of 8 weeks 50% of dermal macrophages are derived from definitive hematopoiesis, i.e. are monocyte-derived. The remaining macrophages are largely derived from cells emerging after E10.5, i.e. fetal liver monocytes, which are seeded in sequential prenatal waves (Kolter et al., 2019, and Hoeffel et al., 2015. Immunity 42, 665-678).

– Methods for fate-mapping of fetal liver monocytes with the necessary sensitivity and specificity, to discriminate their function in local infection, are – to our best knowledge – not available. Moreover, many fate-mapping models rely on tamoxifen treatment, which has substantial immunomodulatory properties impaction on the course of skin infection (Kolter and Henneke, unpublished observation).

– Bone marrow chimeras do not really solve the problem, since dermal macrophages of the recipient mice before infection will already be a mixed population consisting of those derived from fetal liver monocytes and from definitive hematopoesis.

In order to make further progress regarding this point we have sorted and analyzed Ly6C^hi^CX_3_CR1^int^ macrophages, which are monocyte-derived, and Ly6C^lo^ CX_3_CR1^lo^ macrophages, which are in part prenatally seeded for the transcriptional training signature. Notably, that both previously resident and newly monocyte-derived macrophages acquired a memory signature 3 weeks after staphylococcal infection consisting of increased expression of *Cxcl9* and *Stat1* (Figure 5—figure supplement 3). Accordingly, although monocyte-derived macrophages are not essential for the in vivo acquisition of innate memory, the can still be primed to receive a transcriptional memory signature. However, the latter apparently only do so when entering the tissue early after infection, i.e. macrophages derived from monocytes entering the tissue at later stages will extinguish in vivo memory (new model in Figure 7, new Figure 6E).

2) Many experiments were hard to follow (e.g., Figure 6D), and need to be explained in more detail.

We have expanded the description related to Figure 6D and have changed the language to improve clarity in several occasions.

3) Gating strategy needs to be shown (e.g., Figure 1C, D, Figure 4, Figure 5—figure supplement 2A).

The gating strategy is now depicted in Figure 1—figure supplement 1A.

4) Authors should be explicit (e.g., Ly6C^hi^ or Ly6C^lo^ macs) and consistent with macrophage nomenclature throughout study.

We have changed the denomination accordingly to increase consistency as rightfully asked for.

5) Authors should show an analysis of Ly6C^hi^ macs in Figure 1G.

We have added the respective information to Figure 1 as a new panel (new Figure 1H).

6) Figure 3A: It seems like the authors treated primed mice with clodronate and naïve mice with control liposomes. In which case, this cannot be the corresponding FACS analysis for Figure 3B. If so, Figure 3A and B should be repeated by treating primed mice with clodronate or control liposomes, then showing effects on bacterial burden and with the corresponding FACS profiles to indicate depletion of dermal macrophages.

We acknowledge that the description of this experiment was not sufficiently clear. Indeed, the experimental protocol was as follows: Both groups of mice were infected with *S. aureus.* After 2.5 weeks, either clodronate liposomes or control liposomes were injected. Infection was indicated by the red stars in the figure. We apologize for the imprecision and have changed the figure to better explain the procedure.

7) The authors need to show PKH26 labeling specificity.

We acknowledge that this is an important point. Accordingly, we have added new data (Figure 6—figure supplement 2), which show PKH26 labeling in dermal Ly6C^lo^ macrophages, but not in incoming Ly6C^hi^ macrophages after infection.

8) The CFU is strikingly different between Figures 2B and 1C. wt is not included in Figure 2B, leading one to wonder what wt may have looked like. The authors should repeat to include wt in same experiment, or at least the authors need to explain this issue.

We are grateful for the careful analysis of our data. We acknowledge that indeed the bacterial burden in higher in *Ccr2*^-/-^ after infection than in wt mice. However, taking this basic difference into account *Ccr2*^-/-^ mice show a robustly increased bacterial killing after priming, which lasts even longer than the priming effect in wt mice (Figure 6E). In order to accommodate these facts we have changed the text in the Results section relating to Figure 2B.

9) Regarding the half-life of resident dermal macrophages, Figure 1F seems to be pointing to more dermal Ly6C^lo^ macrophage 5 dpi while in Figure 6C, it seems *S. aureus* infection is decreasing the dermal macrophage number. Are the two sets of findings discrepant? Again, an analysis of Ly6C^hi^ macs in Figure 1G would help clarify the issue.

We would like to clarify this. Dermal macrophages are constantly renewed by macrophages differentiating as a progeny of recruited Ly6C^hi^ monocytes. Ly6C appears to be rapidly downregulated under these conditions. This, together with some self-renewal of resident macrophages, keeps the overall macrophage density in the skin stable. The half-life of the macrophages is approximately 5-6 weeks (Figure 6C and D). Notably, in these figures only macrophages resident at the beginning of the experiment are depicted and not those replacing them. In infection, macrophages differentiating as a progeny of recruited Ly6C^hi^ monocytes quantitatively increase dermal Ly6C^lo^ macrophages in the skin at 5dpi (downregulation of Ly6C appears to be fast in vivo during infection). This is depicted in Figure 1F. After infection, previously resident macrophages will be more rapidly replaced by “new” macrophages derived from monocytes. In order to make this point clearer we refer the reviewers to Figure S6E from our previous paper on dermal macrophage origin, which elucidates the macrophage subset redistribution after staphylococcal infection (Kolter et al., 2019).

10) Figure 3A: absolute numbers need to be shown.

We have added the respective data (new data, Figure 3B).